# Topo-AeroVLN:
# Cognitive Topological Mapping for Brain-Inspired Aerial Vision-Language Navigation

## Abstract

Navigating large-scale environments remains a major challenge for autonomous agents. Traditional methods often rely on detailed metric maps, whereas biological systems efficiently navigate using sparse, cognitive topological maps that support high-level reasoning. We present Topo-AeroVLN, a brain-inspired framework enabling unmanned aerial vehicles (UAVs) to perform vision-and-language navigation from a top-down perspective. Our method incrementally constructs a multi-level topological map by abstracting aerial observations into road-bounded regions and internal semantic objects. A dynamic graph update mechanism, combining multimodal embedding similarity with spatial containment, ensures efficient and scalable map construction. Multimodal Large Language Models (MLLMs) align natural language instructions with map vertices, supporting robust language-driven topological planning. Experiments demonstrate strong spatial coverage and navigation performance in complex urban environments. Topo-AeroVLN provides a generalizable, interpretable framework for UAV navigation that adapts to unseen environments without prior maps or extensive retraining.

## 1 Introduction

Brain-inspired architectures are emerging as a powerful paradigm for enabling autonomous agents to operate effectively in complex, dynamic, and unstructured environments. By mimicking biological cognition, these systems tightly couple perception, memory, decision-making, and action—an essential requirement for embodied agents such as drones navigating the real world. In particular, spatial cognition is a core function of biological intelligence: animals construct internal cognitive maps to find paths, remember locations, and adapt to new environments. Neuroscientific studies have uncovered specialized neurons—such as place cells, head direction cells, and grid cells—that encode spatial structures at different levels of abstraction and guide navigation behavior O'keefe & Nadel (1979); Taube (1998); Hafting et al. (2005); Winter et al. (2015). Unlike non-human animals, humans exhibit symbolic abstraction over space, using semantic categories (e.g., "hospital", "residential area") and multi-scale reasoning to flexibly navigate large-scale environments. fMRI evidence suggests that the prefrontal cortex (PFC) supports semantic abstraction while the parahippocampal cortex (PHC) maintains geometric invariance and topological coherence Whitlock et al. (2008); Margulies et al. (2009); Baraduc et al. (2019); Howard et al. (2014). This dual-layer architecture—geometric scaffolding plus semantic overlay—forms the foundation of human-level spatial reasoning and generalization.

Inspired by this architecture, we introduce AeroTopo-VLN, a brain-inspired framework for vision-and-language navigation (VLN) of UAVs in large-scale aerial environments without prior maps. The framework incrementally constructs a cognitive topological map from top-down aerial observations, organizing the environment into road-bounded regions and their internal semantic objects. Given natural language instructions specifying start and goal locations, the system identifies the corresponding regions via visual–textual matching and performs topological planning over the constructed map to generate interpretable navigation paths. Figure 1 provides an overview of this process.

Our main contributions are threefold:

- We introduce a cognitive topological map construction method tailored for aerial VLN, where UAV observations are incrementally organized into a multi-level topological structure.

- The map consists of road-constrained regions and their internal semantic objects. A dynamic update mechanism based on embedding similarity and set-theoretic spatial inclusion enables efficient and scalable graph construction.

- MLLMs are used to generate semantic embeddings and textual descriptions for each map vertex, supporting accurate instruction grounding and shortest-path planning from natural language inputs.

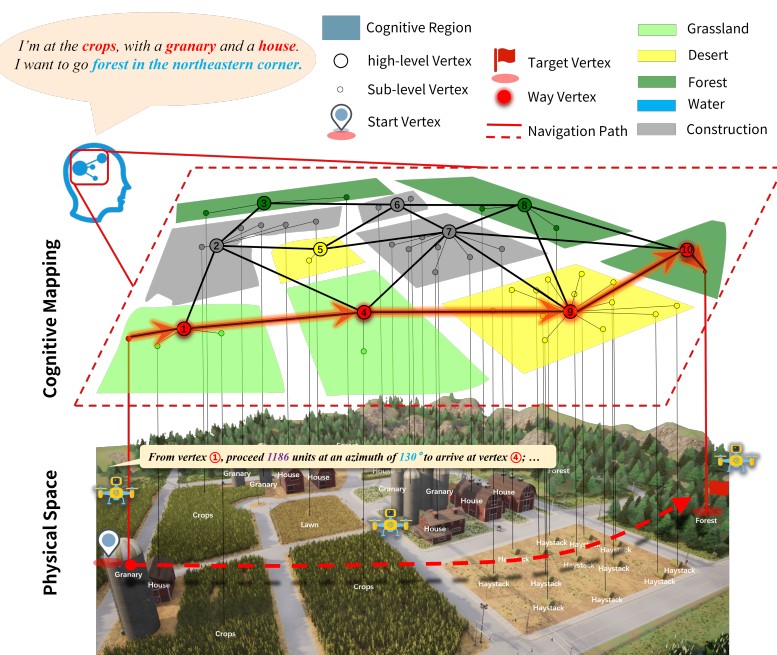

Figure 1: Proposed framework for aerial vision-and-language navigation. It builds a hierarchical cognitive map from aerial views and performs language-guided topological planning.

## 2 RELATED WORKS

### 2.1 COGNITIVE MAPPING AND SPATIAL REPRESENTATIONS

Cognitive maps provide internal structures that enable agents to localize themselves, reason about spatial layouts, and plan routes toward goals. In both neuroscience and robotics, two main paradigms have emerged for representing space: metric (or Euclidean) maps and topological (or relational) maps. Metric maps preserve geometric relationships such as distances and angles, and are often used in SLAM systems to support fine-grained motion planning and localization Wagner (2008); O'Keefe & Nadel (1979); Epstein et al. (2017); Morgan et al. (2011). In contrast, topological maps emphasize the connectivity between places—often ignoring exact geometry—and have proven effective in tasks that require abstraction, generalization, and structural reasoning Warren (2019); Wang et al. (2024a); Eichenbaum (2015).

These two forms of spatial representation reflect different—but complementary—facets of how biological systems handle navigation. For example, place and grid cells in the hippocampal formation encode both metric properties and network-like spatial relations. Empirical findings suggest that animals flexibly switch between geometric and relational representations depending on the task and environment. Cognitive graphs Hartley et al. (2003) extend this notion by incorporating structured semantic knowledge—such as object categories, scene labels, and contextual priors—on top of spatial connectivity. These representations have been widely used in both brain modeling Muller et al.

(1996); Redish & Touretzky (1998); Blum & Abbott (1996) and artificial systems for scene understanding and relational planning.

Despite this rich literature, most existing robotic systems adopt either metric mapping Fiete et al. (2008) or symbolic modeling Eichenbaum & Cohen (2004) in isolation, and approaches that combine both remain rare. This limitation is particularly pronounced in aerial navigation, where observations are noisy, semantics are sparse, and geometry is often ambiguous. To address this, we construct hierarchical cognitive maps that unify geometric structure and semantic abstraction, capturing both spatial connectivity and relational meaning from UAV observations.

### 2.2 Vision-and-Language Navigation in Aerial Environments

VLN tasks aim to guide agents through visual environments based on natural language instructions. Most existing benchmarks, such as Room-to-Room (R2R) Anderson et al. (2018); Wang et al. (2024a;b); He et al. (2024c;a), Touchdown Chen et al. (2019), and CVDN Thomason et al. (2020) focus on ground-level human-centric scenes with predefined graph structures and dense semantic annotations Qiao et al. (2024). These environments assume egocentric perception, short-range geometry, and indoor or street-level affordances that enable fine-grained object grounding and path reasoning.

Recently, VLN has been extended to aerial settings, typically involving low-altitude UAVs navigating within structured urban layouts such as street grids or campus environments Zhang et al. (2025); He et al. (2024b); Gao et al. (2024); Yao et al. (2024). However, these tasks largely preserve road-following or building-aligned paradigms, making the navigable space effectively a constrained aerial variant of ground navigation.

In contrast, high-altitude, top-down VLN introduces challenges that remain largely underexplored Xu et al. (2025); Liu et al. (2024); Sautenkov et al. (2025). From this vantage point, objects appear abstract, semantic boundaries become diffuse, and directional cues are inconsistent. Moreover, explicit paths may be absent, requiring reasoning over regions rather than stepwise trajectories. Such conditions undermine the assumptions of models trained on ground-level datasets and architectures centered on metric maps or waypoint sequences.

To address these limitations, we propose a region-based navigation framework built upon cognitive topological maps derived from aerial observations, enabling agents to capture both spatial connectivity and semantic abstraction in large-scale aerial environments.

## 3 Method

We study vision-language navigation in large-scale aerial environments from a top-down perspective. To address this challenge, we propose a cognition-inspired mapping framework that builds hierarchical semantic representations of the environment, enabling navigation guided by natural language instructions. The overall workflow is shown in Figure 2, and we describe each component in detail below.

### 3.1 Cognitive Map Construction

We employ a UAV equipped with an RGB camera and a semantic segmentation module in the CARLA simulator to explore ground environments from an aerial top-down perspective and collect visual observations. From each frame, high-level vertices are extracted based on the underlying road topology, while within each region, sub-level vertices are identified via the density-based clustering algorithm DBSCAN according to semantic categories. Each vertex is associated with its spatial location (determined by the maximum inscribed circle to guarantee the position lies inside non-convex polygons), image embedding features, textual descriptions, and semantic labels. Notably, semantic labels are assigned only to high-level vertices, while those for sub-level vertices can be incrementally incorporated as the map resolution increases.

The UAV perceives the environment from a top-down perspective and partitions the observed area into regions bounded by roads, which are represented as high-level vertices in the topological graph.

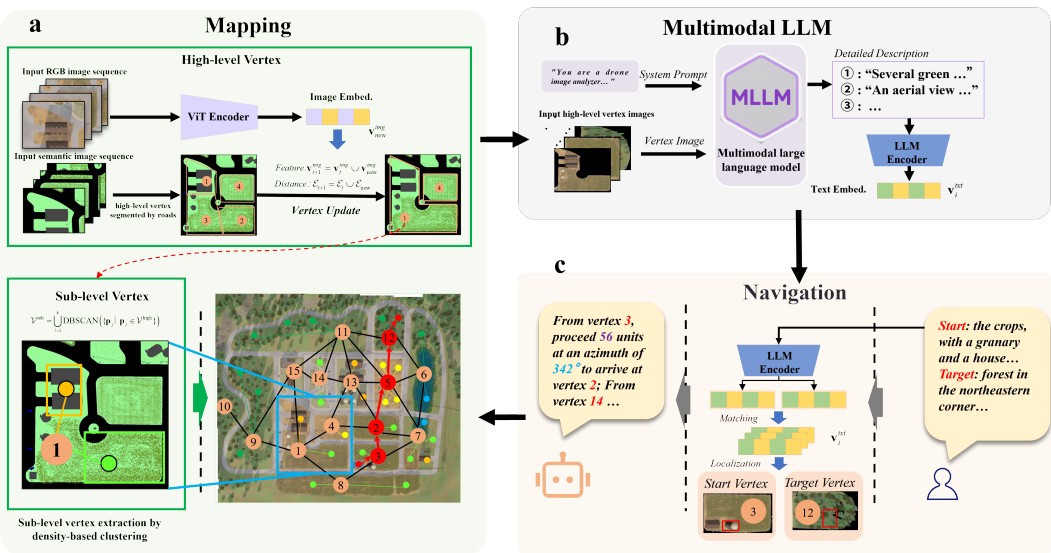

Figure 2: Pipeline of the Topo-AeroVLN. (a) Constructing a cognitive topological map from semantic and RGB inputs. (b) Encoding vertices with text embeddings via an MLLM. (c) Localizing the start vertex from descriptions and planning a navigation path.

A set-theoretic update mechanism is applied, where newly observed data overwrite previous vertex representations while preserving the minimum index order to maintain consistent references.

**Graph Representation.** At timestep $t$, the cognitive topological map is represented as a two-level undirected graph:

$$\mathcal{G}_t = (\mathcal{V}_t, \mathcal{E}_t), \quad \mathcal{V}_t = \mathcal{V}_t^B \cup \mathcal{V}_t^S, \quad \mathcal{E}_t = \mathcal{E}_t^{BB} \tag{1}$$

where $\mathcal{V}_t^B$ are high-level vertices (road-segmented regions) and $\mathcal{V}_t^S$ are sub-level vertices (clustered objects within a region). $\mathcal{E}_t^{BB}$ encodes adjacency among regions according to road connectivity. Each vertex $v$ is described by a tuple $(p_v, \mathbf{v}_v, \Omega_v)$, including its centroid, embedding, and polygonal boundary.

**Vertex Embedding.** For each input frame, we obtain both RGB and semantic segmentation images. The RGB image is encoded by a pretrained ViT to produce an embedding $\mathbf{v}^{\mathrm{img}}$, while the semantic segmentation image provides road masks for region partitioning. Each high-level vertex region $r_i$ is represented by its embedding and geometric attributes:

$$\mathbf{v}_i = \mathbf{v}_i^{\mathrm{img}}, \quad v_i = (p_i, \mathbf{v}_i, \Omega_i) \tag{2}$$

**High-Level Vertex Update.** Given a new frame, each candidate region $r_i$ is compared with existing high-level vertices based on both feature similarity and spatial proximity:

$$\mathrm{sim}(i,j) = \frac{\mathbf{v}_i \cdot \mathbf{v}_j}{|\mathbf{v}i| \cdot |\mathbf{v}j|}, \quad \mathrm{dis}(i,j) = \exp(-\alpha \cdot |p_i - p_j|) \tag{3}$$

$$\tau(i,j) = w\mathrm{sim} \cdot \mathrm{sim}(i,j) + w\mathrm{dis} \cdot \mathrm{dis}(i,j) \tag{4}$$

If $\max_j \tau(i,j) < \theta$, a new high-level vertex is instantiated; otherwise, the region is merged into or updates the most similar existing vertex. Additionally, if the polygonal boundary of one high-level vertex is fully contained within another, the smaller vertex is merged into the larger one to maintain hierarchical consistency.

**Sub-Level Vertex Construction.** After each update, the pixels inside a high-level vertex's region are clustered by density over the semantic segmentation map, producing sub-level vertices $\mathcal{V}_t^S$ (e.g., buildings, vegetation, terrain). These sub-level vertices inherit their parent's region and maintain hierarchical structure.

---

**Algorithm 1** Cognitive Topological Map Update with Set-Based Merging

---

**Require:** New frame RGB image and semantic segmentation image; current map $\mathcal{G}_t = (\mathcal{V}_t^B \cup \mathcal{V}_t^S, \mathcal{E}_t^{BB})$

**Ensure:** Updated map $\mathcal{G}_{t+1}$

1: Extract RGB embeddings $\mathbf{v}_i^{\text{img}}$ via ViT for all regions $r_i$
2: Partition semantic segmentation image into candidate high-level regions $r_i$ with centroids $p_i$ and boundaries $\Omega_i$
3: **for** each candidate region $r_i$ **do**
4:     Compute similarity $\text{sim}(i,j)$ and distance $\text{dis}(i,j)$ with all existing high-level vertices $v_j \in \mathcal{V}_t^B$
5:     Compute combined score $\tau(i,j) = w_{\text{sim}} \cdot \text{sim}(i,j) + w_{\text{dis}} \cdot \text{dis}(i,j)$
6:     **if** $\max_j \tau(i,j) < \theta$ **then**
7:         Instantiate new high-level vertex $v_i = (p_i, \mathbf{v}_i, \Omega_i)$
8:         Add $v_i$ to $\mathcal{V}_{t+1}^B$
9:     **else**
10:         Update or merge $r_i$ into most similar $v_j$
11:     **end if**
12: **end for**
13: **for** each pair of high-level vertices $(v_i, v_j)$ **do**
14:     **if** $\Omega_i \subset \Omega_j$ or $\Omega_j \subset \Omega_i$ **then**
15:         Merge smaller vertex into larger vertex
16:     **end if**
17: **end for**
18: **for** each updated high-level vertex $v_i \in \mathcal{V}_{t+1}^B$ **do**
19:     Perform density clustering on pixels inside $\Omega_i$ from semantic segmentation
20:     Generate sub-level vertices $\mathcal{V}_i^S$ and attach to $v_i$
21: **end for**
22: Construct edges $\mathcal{E}_{t+1}^{BB}$ between high-level vertices whose regions are directly connected by roads

23: Combine $\mathcal{V}_{t+1} = \mathcal{V}_{t+1}^B \cup \bigcup_i \mathcal{V}_i^S$
24: **return** Updated map $\mathcal{G}_{t+1} = (\mathcal{V}_{t+1}, \mathcal{E}_{t+1}^{BB})$

---

**Graph Completion.** Finally, high-level vertices are connected by edges $\mathcal{E}_t^{BB}$ if their corresponding regions are directly linked by road segments, resulting in a complete cognitive topological map. The update procedure is summarized in Algorithm 1.

## 3.2 MLLM-Based Vertex Description

To enable semantic understanding of aerial observations, each high-level vertex in the cognitive topological map is described using a pretrained Multimodal Large Language Model (MLLM). The procedure is as follows:

**High-Level Vertex Encoding.** For each high-level vertex $v \in \mathcal{V}_t$, the corresponding RGB image captured from the top-down UAV view is processed through a prompt-engineered MLLM. The MLLM generates a detailed textual description $d_v$ summarizing the visual and spatial content of the vertex:

$$d_v = \text{MLLM}(\text{RGB}_v) \tag{5}$$

**Text Feature Extraction.** The textual description $d_v$ is then fed into a pretrained LLM Encoder to obtain a $d$-dimensional embedding:

$$\Phi(d_v) \in \mathbb{R}^d \tag{6}$$

This embedding serves as a semantic representation of the vertex, which can later be matched with external language instructions for navigation tasks.

## 3.3 Language-Guided Navigation

Given a natural language instruction specifying source and target locations, the system localizes the corresponding vertices in the cognitive topological map and generates a navigation path.

**Instruction Embedding.** The input instruction

$$q = \text{"I am at } \{\text{desc}_s\}\text{, I want to go to } \{\text{desc}_t\}\text{."} \tag{7}$$

is encoded using the same LLM Encoder to obtain feature vectors:

$$\Phi(\text{desc}_s), \quad \Phi(\text{desc}_t) \in \mathbb{R}^d \tag{8}$$

**Vertex Localization.** The start and target vertices $(v_s, v_t)$ are identified by maximizing cosine similarity between instruction embeddings and vertex embeddings:

$$v_s = \arg\max_{v \in \mathcal{V}_t} \cos\big(\Phi(d_v), \Phi(\text{desc}_s)\big), \tag{9}$$

$$v_t = \arg\max_{v \in \mathcal{V}_t} \cos\big(\Phi(d_v), \Phi(\text{desc}_t)\big) \tag{10}$$

**Topological Path Planning.** Once $(v_s, v_t)$ are determined, a navigation path over the cognitive topological map is generated using classical graph-based planners (e.g., A*) or LLM-based reasoning. The predicted path is a sequence of vertices:

$$\mathcal{P}_{s \rightarrow t} = [v_s, v_1, \ldots, v_k, v_t] \tag{11}$$

which satisfies both topological continuity and semantic relevance.

**Navigation.** The UAV follows the planned path step-by-step, guided by visual-semantic cues corresponding to each vertex. This procedure allows interpretable, language-driven navigation over the constructed cognitive topological map.

Table 1: Comparison of Topological Mapping and Navigation Performance

| METHOD | VERTICES | AVG. VERTEX DEGREE | SPATIAL COVERAGE (%) |
|---|---|---|---|
| Ours | 93 | 8.18 | **81.48** |
| SST Sabag et al. (2025) | 75 | 8.18 | 71.50 |
| Random Walk | 63 | 8.15 | 8.01 |
| RRT | 74 | 8.14 | 67.25 |
| Ours (High resolution) | **181** | **13.91** | 79.37 |

## 3.4 Experimental Configuration

We conduct our experiments in the Town07 map of the CARLA simulator, which offers a complex urban layout suitable for evaluating large-scale topological mapping and language-grounded navigation. The environment spans approximately $300\,\text{m} \times 375\,\text{m}$, providing sufficient spatial diversity for cognitive segmentation and exploration.

In CARLA Town7, we capture a complete global view from a fixed altitude of $150\,\text{m}$, using a field of view (FOV) of $120°$ and a native resolution of $1000 \times 1000$ for both RGB and semantic maps. Flight trajectories are generated by simulating iterative optical reflections to ensure full coverage of the area, which facilitates high-level decomposition and semantic segmentation. To emulate online perception, the global image is initially overlaid with a black mask, and at each sampled trajectory point the corresponding masked region is removed to reveal the UAV's perceptual field. This procedure assumes the availability of an effective image-stitching algorithm, allowing us to focus on the construction of the AeroTopo map.

Each region bounded by roads is abstracted as a vertex in the topological graph, forming the building regions of the cognitive topological map. Within each vertex, high-level vertices represent semantic components such as houses, roads, and vegetation. For every sampled image, we record its RGB content, semantic mask, and precise camera pose, enabling consistent graph construction and multimodal grounding across subsequent processing stages.

## 3.5 MAIN RESULTS AND ANALYSIS

### 3.5.1 COGNITIVE MAPPING PERFORMANCE

**Performance Analysis** To further evaluate the scalability of the mapping process, we compare two resolution settings: $1000 \times 1000$ and $7000 \times 7000$ pixels. As shown in Figure 3(a)-(c), the low-resolution setting significantly reduces runtime and memory usage while preserving high coverage performance. Specifically, memory consumption remains below 1GB, and coverage stays above 80%, suggesting the feasibility of deploying the proposed method on resource-constrained aerial platforms such as UAVs. Moreover, since coverage saturates after around 200 steps and the $1000 \times 1000$ resolution achieves substantially lower per-frame runtime and memory cost than the $7000 \times 7000$ setting, we adopt the $1000 \times 1000$ resolution in all subsequent experiments. We then compare

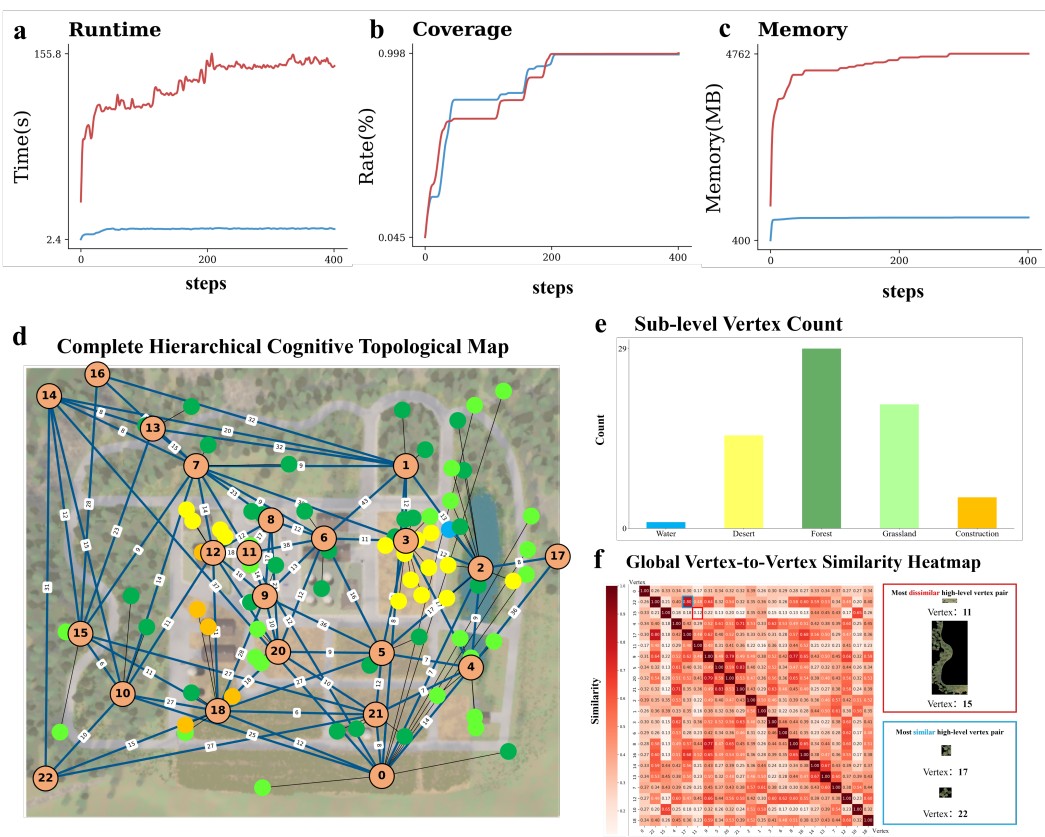

Figure 3: Incremental construction of the cognitive topological map in CARLA Town07. Blue curves correspond to results at $1000 \times 1000$ resolution, while red curves correspond to $7000 \times 7000$ resolution. (**a**) Runtime at different resolutions, (**a**) Spatial coverage at different resolutions, (**c**) Memory usage at different resolutions. (**d**) High-level vertices (regions) and sub-level vertices (objects) over time. (**e**) Distribution of object categories within the map: Water, Desert, Forest, Grassland, and Grassland. (**f**) Pairwise image similarity comparison among different high-level vertices.

our approach against SST Sabag et al. (2025), Random Walk, and RRT under identical resolution settings. As shown in Table 1, our method achieves the highest spatial coverage (81.48%) while maintaining a compact cognitive topological map with only 78 high-level vertices. The average degree of connectivity is 8.18. At a higher resolution, the graph becomes denser with 181 vertices and an increased average degree of 13.91, while coverage remains competitive at 79.37%. These results confirm that our method balances topological richness and semantic abstraction effectively across varying granularities.

Table 2: Semantic grounding accuracy of different MLLMs

| MODEL | R@1 | R@3 | R@5 | COS. |
|---|---|---|---|---|
| GLM-4.1v | 0.318 | 0.545 | 0.682 | 0.726 |
| Moonshot-v1 | 0.182 | 0.454 | 0.455 | 0.682 |
| Qwen-Omni | 0.409 | 0.545 | 0.727 | 0.746 |
| Gemini-2.5 | **0.591** | **0.818** | 0.804 | 0.760 |
| Claude-Sonnet-4 | 0.455 | 0.773 | **0.909** | **0.769** |
| qwen3-vl-plus | 0.364 | 0.682 | 0.773 | 0.734 |
| Ours(7B) | 0.458 | 0.636 | *0.864* | *0.767* |

**Cognitive Topological Map Construction.** Our method incrementally constructs a cognitive topological map from top-down aerial observations. Figure 3(d) illustrates the mapping results in CARLA Town07, consisting of 23 high-level vertices and 70 sub-level vertices. Each high-level vertex represents a spatial region, while sub-level vertices correspond to objects within the region. Distinct colors indicate different object categories. Figure 3(e) summarizes the object distribution: Water, Desert, Forest, Grassland, and Grassland contain 1, 15, 29, 20, and 5 instances, respectively.

To evaluate the representational quality of the cognitive topological map, we analyze the multimodal embeddings of all high-level vertices. Figure 3(f) shows a pairwise similarity heatmap, where red indicates high similarity and white indicates low similarity. Representative examples highlight extreme cases: the least similar pair, vertices 11 and 15, differ substantially in size, geometry, and color, whereas the most similar pair, vertices 17 and 22, are both small forest regions with highly similar appearance, making them difficult to distinguish. These results demonstrate that the learned embeddings are sensitive to both structural and semantic variations across regions.

### 3.5.2 NAVIGATION PERFORMANCE

*Text query "Houses in the L-shaped area"*      *Locate sub-level vertices based on descriptions*

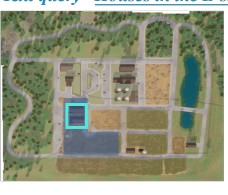 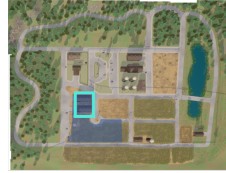 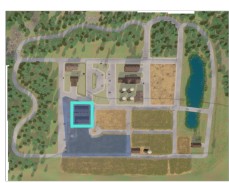 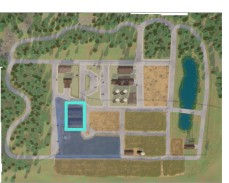

*Text query "storage within building areas"*

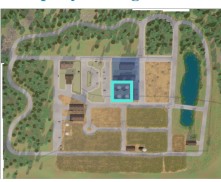 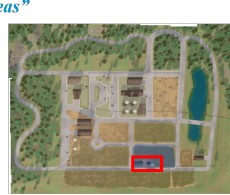 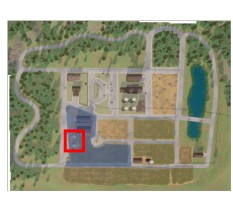 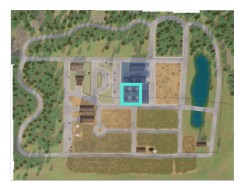

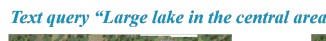

*Text query "Large lake in the central area"*

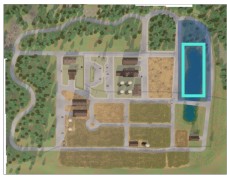 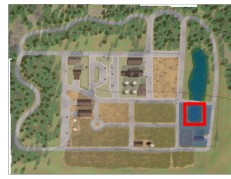 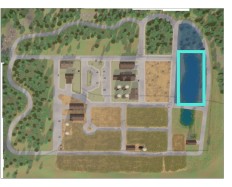 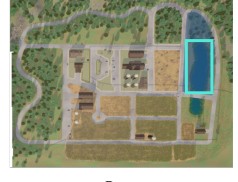

**GT Region of Text**      **GLM-4.1v**      **Gemini-2.5**      **Ours**

Figure 4: Qualitative comparison of query results for high-level and sub-level vertices on a dataset collected from CARLA observations at the same altitude. Blue boxes indicate correctly localized sub-level vertices, while red boxes denote incorrect predictions.

Table 3: MLLM-based path planning metrics on map vertices

| MOEDL | SR | SPL | INV. | INV. RATE | ABS. ERR | REL. ERR | MAX. REL. ERR |
|---|---|---|---|---|---|---|---|
| GLM-4.1v | 0.550 | 0.319 | 85 | 0.368 | 81.7 | 0.591 | 105.500 |
| GPT-4o | 0.853 | 0.649 | 34 | 0.147 | 34.6 | 0.250 | 81.250 |
| Claude-4 | 0.784 | 0.562 | 50 | 0.216 | 40.5 | 0.293 | 72.000 |
| Grok-3 | **0.978** | 0.862 | **5** | **0.022** | 6.2 | 0.045 | **38.750** |
| Moonshot-v1 | 0.957 | 0.823 | 10 | 0.043 | 22.4 | 0.162 | 49.500 |
| Gemini-2.5 | 0.922 | 0.777 | 18 | 0.078 | 19.8 | 0.144 | 63.500 |
| Qwen-Omni | 0.970 | **0.865** | 6 | 0.026 | **4.8** | **0.035** | 255.000 |

## 4 EXPERIMENT

### 4.1 MULTIMODAL LLMS FOR SEMANTIC GROUNDING

We evaluate semantic grounding on the cognitive topological map by identifying the high-level vertex corresponding to a natural language instruction. While ChatGPT-4o can provide human-like vertex descriptions, we perform our experiments using the Qwen3-VL 7B model during map construction, leveraging prompt engineering and relying solely on single-text modality input. This setup reflects a realistic assumption: humans cannot directly transmit imagined regional images from their brain to a robot. The results are summarized in Table 2.

As shown, our Qwen3-VL 7B model achieves competitive performance in semantic grounding despite using only unimodal textual input. The highest R@1 and R@3 scores are obtained by Gemini-2.5 (**0.591** and **0.818**), while the highest R@5 and cosine similarity are achieved by Claude-Sonnet-4 (**0.909** and **0.769**). Our model ranks second in R@5 (*0.864*) and COS (*0.767*), demonstrating that a parameter-efficient model, combined with prompt engineering, can still achieve accurate localization in the cognitive topological map.

We provide brief textual descriptions for selected high-level vertices and simulate UAV image collection in CARLA under a narrow field-of-view, same-altitude setting to form a dataset. The collected images are described using ChatGPT-4o, and these descriptions are then matched against the information of high-level and sub-level vertices in the cognitive topological map for localization. Qualitative results are shown in Figure 4. The results indicate that even state-of-the-art models can make errors in this task when the descriptions of high-level vertices are not explicitly considered.

### 4.2 ABLATION STUDY

**LLM-Based Path Planning Ablation.** We evaluate different LLMs for topological navigation on the cognitive map. Each model generates a path from a high-level start vertex to the target, compared to A*-computed shortest paths using SR, SPL, invalid steps, and absolute/relative errors (Table 3).

Grok-3 achieves the highest SR (0.978) and lowest errors with only 5 invalid steps. Qwen-Omni performs similarly in SPL and relative errors, while GLM-4.1v and Claude-4 show lower success and larger deviations, indicating that model choice strongly affects navigation accuracy.

## 5 CONCLUSION

We presented Topo-AeroVLN, a cognitive topological mapping framework for vision-language navigation from aerial top-down views. Our approach incrementally builds hierarchical maps that integrate geometric structure with semantic abstraction, enabling language grounding without relying on pre-defined metric maps. Experiments demonstrate strong coverage and navigation performance, highlighting the effectiveness of region-based reasoning for UAV navigation. In future work, we aim to deploy the framework on real UAVs for large-scale aerial mapping and explore additional bio-inspired mechanisms to improve system adaptability and robustness.

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

# A APPENDIX

## A.1 ETHICS STATEMENT

This research involved the limited use of LLMs, specifically [e.g., GPT-4, gemini], strictly as an auxiliary tool in two areas:

Manuscript Preparation: To assist with proofreading, checking grammatical errors, and refining the linguistic fluency of the text to improve readability.

Code Development: To assist in modifying and adjusting portions of the code used for data visualization.

It is important to note that all core ideas, scientific conclusions, theoretical analyses, and experimental results are the original work of the authors. The LLM was not used to generate any central scientific insights, data interpretations, or creative content. All outputs generated by the LLM were critically reviewed and verified by the authors, who bear ultimate responsibility for the entire content of this work.

