# OpenReview forum: "Topo-AeroVLN:  Cognitive Topological Mapping for Brain-Inspired Aerial Vision-Language Navigation"
_ICLR.cc/2026/Conference — ICLR 2026 Conference Withdrawn Submission_

### Official Review · Reviewer_4KyY · 2025-10-21

**Soundness:** 2
**Presentation:** 3
**Contribution:** 1
**Rating:** 2
**Confidence:** 5

**Summary:**

This paper introduces a brain-inspired cognitive topological mapping framework for UAV vision-and-language navigation from aerial views. Topo-AeroVLN incrementally constructs a multi-level topological map by abstracting aerial observations into road-bounded regions and semantic objects within those regions. The proposed algorithm updates the map graph dynamically via multimodal embedding similarities and spatial containment, and aligns natural language queries with map vertices using multimodal large language models (MLLMs) to enable language-driven topological path planning. Experiments conducted on the CARLA Town07 simulator demonstrate high spatial coverage with less than 1 GB memory consumption and validate language-driven topological path planning against classical shortest-path algorithms like A*.

**Strengths:**

Combining brain-inspired cognitive mapping with aerial vision-and-language navigation is conceptually novel.

**Weaknesses:**

(1) The practical motivation is unclear. Although the brain-inspired concept is interesting, the aerial-view scenario typically allows access to GPS and classical global path-planning algorithms, raising doubts about the necessity of complex language-driven navigation. A more practical and clearly motivated problem setting is required.

(2) Experiments are limited to a small-scale simulated environment. Given numerous previous studies on aerial vision-and-language navigation like OpenUAV (ICLR25), AerialVLN (ICCV23) and CityNav (arXiv2406.14240), it remains unclear if the proposed approach generalizes or performs competitively on existing benchmarks. While the problem formulation is somewhat novel, experiments of similar scope and scale to previous studies are necessary for clearer validation.

(3) Despite the heavily heuristic design choices in the proposed algorithm (e.g., similarity-distance scoring, containment-based merging, density clustering), the paper lacks a detailed ablation study. Table 3 is described as an ablation but merely compares various LLMs rather than systematically evaluating the contributions of each proposed mapping component.

**Questions:**

(1) What is the practical motivation for using complex language-driven navigation in aerial-view scenarios, given the availability of GPS and traditional global path-planning algorithms?

(2) Can the proposed approach generalize or perform competitively on established benchmarks such as OpenUAV, AerialVLN, or CityNav?

---

> ### Author Response · Authors · 2025-11-28
> **Reply reviewer 4KyY**
>
> # Reply reviewer 4KyY
>
> **Q1_W1_. Practical motivation explanation.**
>
>   Our motivation mainly focus on brain-inspired navigation and GPS-denied navigation.
>
> - **a. brain-inspired navigation**
> Our framework focus on bionic mechanism, such as birds or bat mapping and navigation.
> - **b. GPS-denied navigation**
> It does not rely on GPS and just use vision to perceive the environment, and storing map in the topology structure during the mapping process.
>
> We will add this motivation expression in the introduction.
>
> **Q2_W2_. Can the proposed approach generalize or perform competitively on established benchmarks?**
>
> Our Topo-AeroVLN focuses on buliding a **globle coginative map** with **topo-down view** for **multi-scene** environment. Specifically:
>
> - **a. With a Global Map vs. Without a Global Map:**
>   Benchmarks target small-scale city street settings, where visually similar streets or landmark patterns often lead to ambiguous localization. Our framework is designed for large-scale environments and explicitly constructs a global topological map. Direct comparison would not be meaningful.
>
> - **b. Top-down-view vs. First-view:**
>   Benchmarks assume UAV navigation from a first-view perspective, while our approach is based on a top-down view, leading to fundamentally different environment representation.
>
> - **c. Multi-scene vs. Single-scene:**
>   Benchmarks are all in the city environment or closed environments. Our framework can be applied to various scenarios, such as deserts, forests, lakes, grass, and farmlands.
>
> **W3_: Lack of systematic ablation study on the mapping components.**
> Our mapping pipeline includes:
>
> - *a. Construction of high-level vertices*
> - *b. Clustering of sub-level vertices*
> - *c. Dynamic update mechanism*
>
> These are different from the modules of DeepLearning and are irreplaceable, and based on embedding similarity and spatial inclusion. These are structural constraints required for producing a valid and stable topological map or can merge high-level vertices.

---

### Official Review · Reviewer_nDzp · 2025-10-29

**Soundness:** 2
**Presentation:** 2
**Contribution:** 2
**Rating:** 4
**Confidence:** 3

**Summary:**

The paper introduces a framework that enables UAVs to navigate large-scale environments using a cognitive topological map. The system builds a multi-level map from UAV observations, organizing the environment into road-bound regions and semantic objects, with a dynamic update mechanism for scalable construction. Multimodal language models align natural language instructions with the map, enabling language-guided navigation. Experiments in the CARLA Town07 simulator showed high spatial coverage (81.48%) and effective navigation.

**Strengths:**

1.Cognitive Topological Map Construction: The framework constructs a multi-level topological map using UAV observation data, organizing the environment into road-bounded regions and semantic objects.
2.Dynamic Map Update: A mechanism based on embedding similarity and spatial containment is used to ensure the efficient and scalable construction of the map.
3.Multimodal Large Language Models (MLLMs): These models help align natural language instructions with map vertices, enabling language-driven navigation planning.

**Weaknesses:**

1. The model involves multiple modules, including large language models. Will this result in insufficient UAV inference speed in real-world scenarios?
2. As the UAV gradually builds the cognitive map, could small errors in localization or map updates accumulate over time, leading to reduced accuracy of navigation paths in large-scale environments?
3. why was only CARLA Town07 used for the experiments?

**Questions:**

same as weaknesses.

---

> ### Author Response · Authors · 2025-11-28
> **Reply reviewer nDzp**
>
> # Reply reviewer nDzp
>
> **Q1_W1_: Regarding inference speed and the real-time deployment of MLLMs.**
>
> In our current framework, the mapping process can ensure real-time (**1.5 fps**). Specifically:
>
> - **Topological Mapping Without MLLM:**
>   This relies only on lightweight visual feature extraction and geometric clustering and does not involve the MLLM;
>
> - **Only for Semantic Alignment:**
> LLM needs to be called only when natural language instructions need to be processed, but this stage occurs after mapping and is not important for real-time performance.
>
> **Q2_W2_: Regarding the accumulation of localization/mapping errors over time.**
>
> During the mapping, GPS data was used. Data updates will be made at the next vertex, so errors only exist within the vertex range.
>
> **Q3_W3_: Regarding the experiments being limited to CARLA Town07.**
>
> We selected Town07 primarily because it covers a larger area and includes diverse scene types—such as grasslands, buildings, forests, lakes, farmlands, and haystacks—providing richer features compared to other towns. We will demostrate our work in more towns and real-world in further.

---

### Official Review · Reviewer_VK7C · 2025-10-30

**Soundness:** 2
**Presentation:** 2
**Contribution:** 3
**Rating:** 4
**Confidence:** 3

**Summary:**

This paper presents Topo-AeroVLN, a novel framework for aerial vision-language navigation inspired by cognitive mapping in biological systems. The authors propose a method to incrementally construct hierarchical cognitive topological maps from aerial observations, integrating geometric structure with semantic abstraction. The framework leverages Multimodal Large Language Models (MLLMs) to align natural language instructions with map vertices, enabling robust language-driven navigation. Experiments conducted in the CARLA simulator demonstrate strong spatial coverage and navigation performance, highlighting the potential of this approach for scalable and interpretable UAV navigation without reliance on detailed metric maps.

**Strengths:**

1.	The paper presents a novel method for constructing cognitive topological maps from aerial observations, effectively integrating geometric and semantic information. This approach offers a scalable and interpretable solution for UAV navigation in large-scale environments.
2.	The paper is well-organized with a clear logical flow. The authors conducted thorough ablation studies comparing different MLLMs, which helped identify the most effective model for their framework. This approach demonstrates rigorous experimental design and enhances the credibility of their findings.

**Weaknesses:**

1.Mismatch Between Title and Content: The title of the paper focuses on Vision-Language Navigation (VLN), but the proposed framework appears to be more aligned with a semantic-enriched topological mapping approach rather than the conventional VLN task setting. The paper consider navigation in high-altitude aerial environments, where data collection and topological map generation suggest that the drone operates at a height where obstacles are not encountered. Given the typical VLN task design, where the prompt only requires navigating from a start point to an end point without additional obstacles, the drone’s navigation path should ideally be a direct route from start to finish. The reliance on a topological map to guide the path seems unnecessary in this context, deviating from the standard VLN task assumptions.

2.Writing and Presentation: The authors are encouraged to revise the manuscript to enhance clarity and readability, particularly in the presentation of formulas and figures. For example:
- In Formula (4), the symbols for weights (sim and dis) and the function name (sim) appear at the same level, which may cause confusion for readers. The authors should consider clarifying these symbols or reformatting them, such as using subscripts for weights (e.g. ω_sim).
- In Figure 3(a-c), the two curves are not accompanied by a legend, and the text does not directly explain their meanings. Readers are left to infer the significance of these curves, which can increase the difficulty of understanding the paper. The authors should add a clear legend to the figure and provide a more explicit explanation in the text.

**Questions:**

See the weaknesses section.

---

> ### Author Response · Authors · 2025-11-28
> **Reply reviewer VK7C**
>
> # Reply reviewer VK7C
>
> **Q1_W1_: There is a mismatch between the title and the content.**
>
> Our research is a VLN task, but compared with the traditional VLN task, We have done more work to conduct mapping before navigation. Cognitive Map is similar to the prompt input of MLLM. This can be more in line with the cognitive behavior of the human brain.
>
> **Q2_W2_: The writing and presentation require improvement.**
>
> We thank you for pointing out the specific technical details. In the revised version, we have made the following improvements:
> Clarification of symbols in Equation (4):
> All subplots now include clear legends, explicitly indicating which curves correspond to the “1000×1000 resolution” and the “7000×7000 resolution.”
> Direct explanation in the main text (Section 3.5.1):
> We have added statements clarifying the meaning of the curves, for example: “The blue curve represents runtime/coverage/memory consumption under the low-resolution setting, while the red curve corresponds to the high-resolution setting.”
>
> We also have thoroughly checked figure references, terminology consistency, and mathematical expressions throughout the manuscript to improve overall readability and rigor.

---

### Official Review · Reviewer_hpHs · 2025-10-31

**Soundness:** 3
**Presentation:** 3
**Contribution:** 3
**Rating:** 6
**Confidence:** 3

**Summary:**

This paper presents Topo-AeroVLN, a brain-inspired framework for aerial vision-and-language navigation (VLN). It builds a two-level cognitive topological map—road-bounded regions (high-level nodes) and clustered semantic objects (sub-level nodes)—and updates it via embedding similarity + polygon containment. Each region is described by an MLLM-generated caption and aligned to language queries for topological navigation. Experiments in CARLA Town07 show >80% coverage and competitive retrieval and path-planning accuracy.

**Strengths:**

- Addresses an underexplored high-altitude aerial VLN scenario with sparse semantics.
- Proposes a region-based topological representation that supports scalable and interpretable navigation.
- The set-theoretic merging rule and language grounding via MLLMs are conceptually neat and well-integrated.
- Experiments include several MLLMs for grounding and planning, offering practical insights.

**Weaknesses:**

- No direct comparison with existing aerial VLN baselines (AerialVLN, CityNavAgent, See-Point-Fly).
- Evaluation assumes perfect segmentation and stitching, lacking robustness analysis for real UAV data.
- Ablations only test different LLMs, not the mapping components (e.g., merging, sub-level hierarchy).

**Questions:**

- Can the authors benchmark against existing aerial VLN methods with SR/SPL metrics?
- How robust is the mapping to segmentation errors or partial occlusion?
- Are there plans to validate on real UAV data or release the dataset?

---

> ### Author Response · Authors · 2025-11-28
> **Reply reviewer hpHs**
>
> # Reply reviewer hpHs
>
> **Q1_W1_: No direct comparison with existing aerial VLN benchmark.**
>
> Benchmarks only have the function of navigation, and the global maps, perspectives and scenarios of our framework and benchmarks are all different. Specifically:
>
> - **a. With a Global Map vs. Without a Global Map:**
>   Benchmarks target small-scale city street settings, where visually similar streets or landmark patterns often lead to ambiguous localization. Our framework is designed for large-scale environments and explicitly constructs a global topological map. Direct comparison would not be meaningful.
>
> - **b. Top-down-view vs. First-view:**
>   Benchmarks assume UAV navigation from a first-view perspective, while our approach is based on a top-down view, leading to fundamentally different environment representation.
>
> - **c. Multi-scene vs. Single-scene:**
>   Benchmarks are all in the city environment or closed environments. Our framework can be applied to various scenarios, such as deserts, forests, lakes, grass, and farmlands.
>
> **Q2_W2_: How robust is the mapping to segmentation errors or partial occlusion?**
>
> Local-scale errors may occur, but these are acceptable and tolerable in large-scale environments.  We have supplemented additional experiments in the table below, which will be presented in the form of charts in the main text.  We conducted the Error statistics of mapping based on the diagonal length of the map. **Error range** denotes the normalized error, computed as the raw localization error divided by the diagonal length of the map. **Proportion** is the proportion of vertex to the total number of vertices.
>
> | Error range | Number of high-level vertex | Proportion |
> | :---------: | :-------------------------: | :--------: |
> | [0.0, 0.1)  |             14              |   30.4%    |
> | [0.1, 0.2)  |              9              |   19.6%    |
> | [0.2, 0.3)  |              6              |   13.0%    |
> | [0.3, 0.4)  |              9              |   21.7%    |
> | [0.4, 0.5)  |              4              |    8.7%    |
> | [0.5, 0.6)  |              3              |    6.5%    |
>
> You pointed out that our evaluation assumes perfect segmentation and stitching. We have replaced the input images with real-time simulated UAV local observations from the CARLA instead of using global images. We have also introduced low-resolution conditions to evaluate the robustness of our method.
>
> **Q3_: Are there plans to validate on real UAV data or release the dataset?**
>
> Yes, we will release the dataset in further.
>
> Currently, our framework is still a prototype system. We focus on presenting a new framework that integrates both mapping and navigation. At this stage, we conduct thorough evaluations in a simulation environment, which is a necessary step before deployment in real-world settings. Our Topo-AeroVLN also features **low power consumption**,  making it suitable for real-time deployment on UAVs.
>
> **W3_: Ablations only test different LLMs, not the mapping components.**
>
> Our mapping pipeline includes:
>
> - *a. Construction of high-level vertices*
> - *b. Clustering of sub-level vertices*
> - *c. Dynamic update mechanism*
>
> These are different from the modules of DeepLearning and are irreplaceable, and based on embedding similarity and spatial inclusion. These are structural constraints required for producing a valid and stable topological map or can merge high-level vertices.

---

### Author Response · Authors · 2025-12-01
**General response to the reviewers and AC**

Our work aims to build a brain-inspired Vision-and-Language Navigation (VLN) framework for aerial agents, validated in simulation with the goal of enabling future deployment on real UAVs—rather than pursuing incremental performance gains.

We have addressed all reviewers’ concerns regarding research motivation, comparison with benchmarks, and ablation studies of mapping components. A summary is provided below:

**1. Research Motivation.**

 Our motivation mainly focus on brain-inspired navigation and GPS-denied navigation.

 **a. brain-inspired navigation.**

  Our framework focus on bionic mechanism, such as birds or bat mapping and navigation.

  **b. GPS-denied navigation.**

  It does not rely on GPS and just use vision to perceive the environment, and storing map in the topology structure during the mapping process.

**2. Comparison with Benchmarks.**

Our Topo-AeroVLN focuses on buliding a globle coginative map with topo-down view for multi-scene environment. Specifically:

- **a. With a Global Map vs. Without a Global Map.**

 Benchmarks target small-scale city street settings, where visually similar streets or landmark patterns often lead to ambiguous localization. Our framework is designed for large-scale environments and explicitly constructs a global topological map.

  - **b. Top-down-view vs. First-view.**

 Benchmarks assume UAV navigation from a first-view perspective, while our approach is based on a top-down view, leading to fundamentally different environment representation.

  - **c. Multi-scene vs. Single-scene.**

 Benchmarks are all in the city environment or closed environments. Our framework can be applied to various scenarios, such as deserts, forests, lakes, grass, and farmlands.

**3. Ablation Study of Mapping Components.**

  Our mapping pipeline includes:

  *a. Construction of high-level vertices,*

  *b. Clustering of sub-level vertices,*

  *c. Dynamic update mechanism.*

  These are different from the modules of DeepLearning and are irreplaceable, and based on embedding similarity and spatial inclusion. These are structural constraints required for producing a valid and stable topological map or can merge high-level vertices.

---

### Note · Authors · 2026-01-26

I have read and agree with the venue's withdrawal policy on behalf of myself and my co-authors.

---

### Meta-Review · Area_Chair_m6HY · 2025-12-25

**Summary:**

The paper proposes Topo-AeroVLN for aerial vision–language navigation that constructs a multi-level topological map and leverages MLLMs for planning.

The initial reviewer scores are 6, 4, 4, and 2. Notably, the reviewer assigning a score of 2 expressed high confidence (5), while the other reviewers reported moderate confidence (3).

The primary concerns include: (1) evaluation limited to a simulated environment (nDzp, 4KyY), (2) the absence of comparisons on established VLN benchmarks and with existing aerial VLN baselines (hpHs, 4KyY),  and (3) missing ablation studies of the proposed mapping components (hpHs, 4KyY).

In the rebuttal, the authors did not adequately address these issues. Specifically, they argued that the mapping components cannot be ablated and did not provide additional experimental comparisons.

Given the limited empirical validation and the lack of substantive responses to the reviewers’ core concerns, the AC recommends rejecting the paper.

**Reviewer Concerns:**

See above.

**Reviewer Scores:**

The reviewers can lower the score.

---

### Decision · Program_Chairs · 2026-01-26

Reject